# Knowledge and practices of modern contraceptives among religious minority (Muslim) women: A cross-sectional study from Southern Nepal

Dipendra Singh Thakuri[1]*, Yamuna K. C. Singh[2], Rajendra Karkee[3], Resham B. Khatri[4,5]

1 Save the Children, Nepal Country Office, Kathmandu, Nepal, 2 Rapti Academy of Health Sciences, Ghorahi Dang, Nepal, 3 School of Public Health and Community Medicine, BP Koirala Institute of Health Sciences, Dharan, Nepal, 4 School of Public Health, Faculty of Medicine, University of Queensland, Brisbane, Australia, 5 Health Social Science and Development Research Institute, Kathmandu, Nepal

* dipendrathakuri@gmail.com

**Data Availability Statement:** All relevant data are within the paper and its Supporting Information files.

## Abstract

### Background

Uptake of family planning (FP) services could prevent many unwanted pregnancies, and unsafe abortions and avert maternal deaths. However, women, especially from ethnic and religious minorities, have a low practice of contraceptives in Nepal. This study examined the knowledge and practices of modern contraceptive methods among Muslim women in Nepal.

### Methods

A cross-sectional study was conducted among 400 Muslim women in the Khajura Rural Municipality of Banke district. Data were collected using face to face structured interviews. Two outcome variables included i) knowledge of and ii) practices of contraceptives. Knowledge and practice scores were estimated using the list of questions. Using median as a cut-off point, scores were categorised into two categories for each outcome variable (e.g., good knowledge and poor knowledge). Independent variables were several sociodemographic factors. The study employed logistic regression analysis, and odds ratios (OR) were reported with 95% confidence intervals (CIs) at a significance level of p<0.05 (two-tailed).

### Results

Almost two-thirds (69.2%) of respondents had good knowledge of modern contraceptive methods, but only 47.3% practised these methods. Women of nuclear family (adjusted odds ratio (aOR) = 0.60; 95% CI: 0.38,0.95), and who work in agricultural sector (aOR = 0.38; 95% CI: 0.22, 0.64) were less likely to have good knowledge on modern contraceptives. Women with primary (aOR = 2.59; 95% CI: 1.43, 4.72), secondary and above education (aOR = 4.41; 95% CI:2.02,9.63), women with good knowledge of modern contraceptives

**Funding:** The authors received no specific funding for this work.

**Competing interests:** The authors have declared that no completing interests exist.

**Abbreviations:** FP, Family Planning; FPAN, Family Planning Association of Nepal; NFCC, Nepal Fertility Care Center; IUCD, Intrauterine Contraceptive Device; SBCC, Social Behavior Change Communication; HW, Health Workers; FCHV, Female Community Health Volunteer; NHSS, Nepal Health Sector Strategy; MOHP, Ministry of Health and Population; PHCC, Primary Health Care Center.

(aOR = 2.73; 95% CI: 1.66, 4.51), who ever visited a health facility for FP counselling (aOR = 4.40; 95% CI: 2.58, 7.50) had higher odds of modern contraceptives practices.

## Conclusion

Muslim women had low use of modern contraceptive methods despite having satisfactory knowledge about them. There is a need for more equitable and focused high-quality FP practices. Targeted interventions are needed to increase the knowledge and practices of contraceptives in the Muslim community. The study highlights the need to target FP interventions among socially disadvantaged women, those living in a nuclear family, and those with poor knowledge of modern contraceptives.

## Introduction

Family planning (FP) is one of the high-impact interventions that prevent unintended pregnancies, unsafe abortions, reduce high-risk births, avert maternal and neonatal deaths, and protect women's and children's health [1–4]. Despite multiple benefits, many women needing FP methods cannot access the FP services. This unmet need for FP results in approximately 539,000 annual unintended pregnancies in Nepal [5, 6]. These unintended pregnancies can pose serious health risks to mothers and their newborns, including deaths [7]. Maternal morbidity and mortality risks are also high among poor, rural women facing many barriers to accessing FP services in Nepal [5, 8, 9]. One in 200 women dies from pregnancy-or delivery-related causes in their lifetime in Nepal [10].

Nepal made considerable progress in health services access and improved maternal and child health services coverage over the last three decades [11, 12]. However, the FP program has poor performance and has low and stagnant progress in the contraceptive prevalence rate (CPR) [13]. The Nepal Demographic Health Survey (NDHS) 2016 [12] revealed that CPR for modern contraceptive methods in Nepal was 43%, with 24% unmet needs. In addition, women from the poorest households, living in remote areas, disadvantaged ethnicities, religious minorities, and those without education had poor knowledge and the lowest practice of contraceptive methods [14].

Nepali Muslims are recognised as one of the most marginalised and disadvantaged communities [15], consisting of 4.4% of the total population [16]. Most of them live in the Terai districts. Muslims are economically, socially, educationally, and politically backward and deprived of various facilities, including health services [15]. In 2011, the poverty incidence for the Muslim population was 20.2%, and the adult literacy rate was 43.5%, compared with 25.2% and 40.43% for the general population, respectively [16]. Muslims ranked at the bottom of the Human Development Index (HDI) (HDI score: 0.41) [17]. In Nepal, Muslims have low access to and practice of family planning with a high unmet need for FP services, including other health services [16, 18]. Muslim women have low CPR (25.4%), high unmet need (37%) for modern contraceptive methods, high fertility and large family size in Nepal [16, 19]. In Nepal, the total fertility rate has increased from 4.6 (2006) to 4.9 (2011) in Muslim populations [16]. Muslim groups had high unintended pregnancies leading to the highest maternal mortality ratio (318 per 100000 live births) in Nepal [20], which suggests the need for quality FP services delivery and utilisation among Muslims. Better use of family planning could reduce many of these mistimed and unplanned pregnancies. At the same time, it could reduce the number of unsafe abortions and the mortality related to childbirth [21].

Several factors have contributed to poor progress in practices of contraceptive methods, including poor access to contraceptive methods, lack of contraceptive methods in health facilities [22], poor uptake due to perceived side effects, poor knowledge of contraceptive methods, opposition from family members, psychological factors, lack of proper counselling services on contraceptive methods and religious and cultural beliefs and value system [18, 23]. In addition, behavioural norms prevailing in Muslim society may affect Muslim women's access to and utilisation of family planning services [24]. For example, a common concern in Muslim communities is that FP is deemed a Western ideology and a conspiracy to reduce the Muslim population [19]. Previous studies reported that some Muslims believe that using family planning services will result in divine retribution or that the number of children they should have is 'God's business,' and that parents should not try to change God's will [16, 25].

Evidence showed that knowledge and attitude contributed to modern contraceptive methods [26]. In addition, other socioeconomic and demographic factors were also identified as determinants of contraceptive methods, such as women's age, education, number and sex of children, occupation, and access to a health facility [3, 18]. However, limited evidence is available on the status of knowledge and practices of modern contraceptive methods and their associated determinants among Muslim women in Nepal.

Therefore, we aimed to assess the existing family planning knowledge and practice among Muslim women and identify the factors influencing access to and uptake of modern family planning services in Mid-western Nepal. The findings of this study could inform policymakers and program managers to design contextual policies and programmatic strategies for universal coverage of contraceptive methods among the Muslim population.

## Policy and services delivery context of family planning program in Nepal

The FP program is Nepal's oldest public health program [19], and services are available at the community level through Female Community Health Volunteers (FCHVs). Nepal's health policy 2019 and strategies also emphasised the family planning program and ensuring quality family planning services. Current periodic strategies and plans, such as Nepal Health Sector Strategy (NHSS) 2015–2020 [27], and the Population Perspective Plan (2010–2031) [28], have highlighted family planning as the major component of the Safe Motherhood Initiative in Nepal [29]. The family planning Costed Implementation Plan 2015–2021 has also highlighted the cost and implementation strategies [30]. However, these policies and program approaches are implemented one-size-fits-all [19]. There have not been focused and context-specific implementation strategies to recognise religious and cultural considerations for addressing FP needs of marginalised populations.

In Nepal, modern contraceptive services provided from different outlets ranging from community to tertiary level (Fig 1). Services outlets include community clinics, health posts, static health clinics, and mobile health camps from public, private, and private non-profit sector health institutions. In addition, several short-term modern contraceptives are available at peripheral facilities. In contrast, long-term modern contraceptives are being provided in health posts (HP), primary health care centers (PHCC) and hospitals [31].

## Methods

### Study design and setting

A community-based cross-sectional study was carried out between June and September 2019 in Khajura Rural Municipality of Banke district. The study population was married Muslim

**Fig 1. Types and delivery outlets of modern contraceptives in Nepal.** Source: Developed by authors based on information obtained from Nepal's annual health report 2019 [31].

women with reproductive ages of 15 to 49 years. We selected Khajura Rural Municipality purposively. More than one in four (26.7%) people in this municipality belong to Muslim backgrounds [32]. Khajura Rural Municipality has 50,961 residents from 10,288 households (Female: 27,457 and 19,397 aged 15 to 49 years) [32]. Four wards (of eight wards) of the municipality were selected randomly for the household survey. A Ward is the lowest administrative unit in Nepal. An estimated 1,750 Muslim married women of reproductive age (MWRA) were living in those selected wards [33].

## Sampling and participants selection

This study's sampling frame was married Muslim women aged 15–49 years. The sampling frame of Muslim MWRA was obtained from the selected ward office. Sample size was calculated using formula $N = Z^2pq/d^2$ where [Z = 1.96, p = 0.44 q = 0.56, d = 0.05] and 44% prevalence rate [34]. We determined 379 as the minimal sample size. Considering a non-response rate of 5% [35], a sample of 400 Muslim women were interviewed among 1,750 Muslim MWRA. We selected participants through a systematic random sampling method. The first women were selected randomly, and then every fourth (having a gap of three) women were selected for the interview. If there was more than one MWRA in the family, the youngest women were included in the study. Likewise, the adjoining households were recruited if the participants were unavailable in the selected households.

## Study variables

Based on previous studies in Nepal and elsewhere [19, 36, 37], explanatory variables were basic socioeconomic and demographic variables. Demographic variables were respondent's age ($\leq$18 years,19–29 years and $\geq$30 years), parity (0 to 2 and $\geq$3), respondent's family type (nuclear and joint family) [38]. Socioeconomic variables were respondent's education (illiterate, primary education, and secondary and above) [39], respondent's occupation (agriculture, daily wage workers and housewives). Similarly, family monthly income ($\leq$20000 NRs and >20000 NRs (130 Nepalese Rupees = 1 USD, 2022)). Additionally, access to family planning service variable included: ever visited a health facility for family planning counselling (yes/no). Knowledge of modern contraceptive methods was also included as the independent variable for practices of modern contraceptive methods.

## Outcome variables

Two outcome variables were included: knowledge on modern contraceptive methods (good and poor knowledge), and practices of modern contraceptive methods (yes or no). Knowledge of modern contraceptive methods was created using ten questions about modern contraceptives. Each question's response was coded as "1" for "yes" and "0" for "no". The possible knowledge score ranges between a minimum 0 and maximum 10. Next, the median score of the knowledge was calculated. Using the median as a cut-off-point, we categorised knowledge level into "Good" (> = median score) and "Poor" (<median score) [7, 40].

Women were asked if they had used modern contraceptive methods in the last six months before this survey and coded their responses as 'yes' or 'no' to assess the practice of modern contraceptive methods.

## Data collection tools and techniques

A questionnaire on knowledge and practice of modern contraceptive methods was adopted from the previous studies [19, 34, 41] and a survey [12]. The structured questionnaire was first developed in English, and then translated into Nepali and the local language (Awadhi). The second author YKCS translated the English version of the tool into Nepali and the local language (Awadhi) with the support of a professional translator. It was pretested among 20 women aged 15–49 years in an adjoining ward to refine it. Necessary adjustments were made, including in the flow of question patterns and language style. The local language was used in data collection. A face-to-face interview was conducted in the participant's households. The interview was carried out in a separate area of the participants' households to ensure confidentiality. Participation was voluntary, and none approached respondents who refused to be interviewed. Data were collected by local enumerators consisting of three females. The enumerators were the local Muslim community. They were recruited based on their educational background, local language knowledge, and prior data collection experience. The two days of training were provided to the enumerators about the study purpose, methodologies, tools, and techniques before preceding the actual data collection. All the data collection-related field activities were closely supervised and monitored by the second author (YKCS).

## Data analysis

Data analysis was performed using SPSS version 25.0 (SPSS Inc., Chicago, IL). The collected data were entered, coded, and cross-checked to ensure consistency. Descriptive analyses were employed and reported as frequencies and proportions. The Chi-square test was conducted to assess the association between independent and outcome variables. Binomial logistic

regression was examined to identify the determinants of knowledge and practices of modern contraceptive methods. Odds ratio with 95% confidence interval (CIs) were reported. The significance level was set at $p < 0.05$ (two-tailed).

### Ethical approval

Ethical approval was obtained from this study's ethical review board of Nepal Health Research Council and the educational and administrative ethical committee, faculty of Nursing and Medical College of Xi'an Jiaotong University, China. Before collecting data, written permission was obtained from the local administrative authority Khajura Rural Municipality of Banke district. Before the interview, enumerators and the second author (YKCS) met Muslim religious leaders, shared the study's objective, and obtained permission to meet and collect data from their community. Verbal informed consent was obtained from participants before conducting the interview. The respondent's participation was voluntary, and the respondents had the right to refuse the interview process.

## Results

Table 1 shows the distribution of respondents accordingly to sociodemographic characteristics, the prevalence of knowledge and the use of modern contraceptive methods. Nearly half (46%) of the respondents were between 19 and 29 years. The mean age of respondents was 29 (±8.74 SD) years. Over three-quarters of respondents (78.5%) had up to 2 living children, and almost half (48%) of respondents had primary level education. Approximately half (49.5%) of respondents were housewives, while 37% of respondent's husbands were involved in agriculture. Over 6 in 10 respondents had >20000 NRs family monthly income. Almost two-thirds (69.2%) of respondents had good knowledge of modern contraceptive methods, and 47.3% used modern contraceptive methods. Overall, the mean knowledge score of FP was 6.9 (±1.18), with minimum knowledge scores of 3 and maximum scores of 10, and the median knowledge score was 7. Injectable (43.4%) was the most used modern contraceptive, and an implant (3.7%) was the least used. Additionally, over 7 in 10 (71%) visited a health facility for family planning counselling (Table 1).

Table 2 shows the descriptive findings of knowledge regarding modern contraceptive methods. The majority of respondents (89%) heard about family planning. However, almost half (47.3%) of respondents didn't know that using both a condom and oral contraceptives is effective. Over 7 in 10 (71%) respondents knew that women who use injectables must get an injection every three months. Almost seven in ten (69%) women knew that using contraceptives prevents unwanted pregnancies. However, over four in ten (44%) women didn't know about contraceptive pills' common side effects, as shown in Table 2.

Table 3 depicts the different sociodemographic variables, knowledge, and practice of modern contraceptive methods. Nearly three fourth (72.5%) of respondents from a joint family had good knowledge of modern contraceptive methods. Over half (56.6%) of respondents belonging to the nuclear family practised modern contraceptive methods. Respondents with secondary and above education reported greater (56.0%) use of modern contraceptive methods. More than half (51.6%) of women who had good knowledge of modern contraceptive methods used modern contraceptives. Six in ten (60.3%) respondents who visited a health facility for FP counselling have used modern contraceptive methods (Table 3).

Table 4 illustrates the determinants of knowledge on contraceptive methods. Women who belonged to the nuclear family (aOR = 0.598; 95% CI: 0.38,0.95) had lower odds of knowing modern contraceptive methods than those in the joint family. Women who involve in

**Table 1. Background characteristics and modern contraceptive methods, knowledge, and practices of Muslim women (N = 400) in Nepal.**

| Variables | Category | Frequency | Percentage |
|---|---|---|---|
| Age of women | ≤18 Years | 34 | 8.5 |
| | 19–29 Years | 184 | 46.0 |
| | ≥30 Years | 182 | 45.5 |
| Parity | 0 to 2 | 314 | 78.5 |
| | 3 or above | 86 | 21.5 |
| Family types | Nuclear | 145 | 36.3 |
| | Joint | 255 | 63.7 |
| Women's education | Illiterate | 108 | 27.0 |
| | Primary | 192 | 48.0 |
| | Secondary & above | 100 | 25.0 |
| Women's occupation | Agriculture | 158 | 39.5 |
| | Daily wages worker | 44 | 11.0 |
| | Housewives | 198 | 49.5 |
| Husband's occupation | Agriculture | 148 | 37.0 |
| | Business and service | 50 | 12.5 |
| | Daily wages worker | 82 | 20.5 |
| | Foreign migrant worker | 120 | 30.0 |
| Family income (NRs) | ≤20000 | 158 | 39.5 |
| | >20000 | 242 | 60.5 |
| Knowledge of modern contraceptive methods | Poor knowledge | 123 | 30.8 |
| | Good knowledge | 277 | 69.2 |
| Practices of modern contraceptive methods | Yes | 189 | 47.3 |
| | No | 211 | 52.7 |
| Contraceptive practices (n = 189) | Condom | 33 | 17.5 |
| | Oral contraceptive | 49 | 25.9 |
| | Injectable | 82 | 43.4 |
| | Implant | 7 | 3.7 |
| | Intrauterine contraceptive Device (IUCD) | 10 | 5.3 |
| | Female sterilisation | 8 | 4.2 |
| Ever visited a HF for FP counselling | Yes | 284 | 71.0 |
| | No | 116 | 29.0 |

agricultural sector (aOR = 0.379; 95% CI: 0.22, 0.64) were less likely to be aware of modern contraceptive methods than housewives (Table 4).

Table 5 demonstrates the determinants of practice of modern contraceptive methods. Women with primary (aOR = 2.59; 95% CI: 1.43, 4.72), secondary and above education (aOR = 4.41; 95% CI:2.02,9.63) had significantly higher odds of practices of modern contraceptive methods compared to illiterate women. Women living in a nuclear family (aOR = 2.24; 95% CI:1.40,3.59) had more than two-fold higher odds of modern contraceptive practices than their counterparts. Additionally, the practices of modern contraceptives were significantly higher among women in the business and service sector (aOR = 2.55; 95% CI:1.17,5.56) compared to agriculture. Women having good knowledge of modern contraceptive methods (aOR = 2.73; 95% CI: 1.66, 4.51) and women who ever visited a health facility for FP counselling (aOR = 4.40; 95% CI: 2.58, 7.50) were more likely to practice modern contraceptive methods compared to those who had poor knowledge and those who have not visited a health facility for FP counselling respectively (Table 5).

**Table 2. Descriptive findings of knowledge related to modern contraceptives among Muslim Women, Banke, Nepal, 2019 (N = 400).**

| Variables | Yes | No |
|---|---|---|
| | Number (%) | Number (%) |
| | n (%) | n (%) |
| 1. Have you ever heard about FP? | 356 (89%) | 44 (11.0%) |
| 2. Does female sterilisation avoid pregnancy? | 288 (72%) | 112 (28.0%) |
| 3. Do oral contraceptive pills guarantee 100% protection? | 269 (67.2%) | 131(32.8%) |
| 4. Are women using the birth control injectables to get an injection every three months? | 284 (71.0%) | 116 (29.0%) |
| 5. Do the use of both a condom and oral contraceptives considered to be very effective contraceptives? | 211(52.8%) | 189 (47.2%) |
| 6. Does use of contraceptives prevent unwanted pregnancies? | 276 (69.0%) | 124 (31.0%) |
| 7. Are contraceptive methods appropriate to space childbirth? | 268 (67.0%) | 132 (33.0%) |
| 8. Does condom provide dual protection (Prevents STI/HIV and unplanned pregnancies) | 286 (71.5%) | 114 (28.5%) |
| 9. Do common side effects of contraceptive pills include mood swings and weight gain? | 224 (56%) | 174 (44%) |
| 10. Does health education important for women who want to use contraceptives? | 266 (66.5%) | 134 (33.5%) |

## Discussion

The current study assessed the knowledge and practice of modern contraceptives among Muslim women. Most Muslim women had relatively good knowledge and poor practice of modern contraceptives. Knowledge of modern contraceptive methods was low among the women working in agriculture and living in nuclear families. The practice of modern contraceptives was poor among women with no education, husbands working in the agriculture sector, women having poor knowledge on modern contraceptive methods, and who have not visited a health facility for family planning counselling.

This study revealed that 69% of women had good knowledge of modern contraceptive methods. Past studies reported mixed results on knowledge of modern contraceptive methods in Nepal. For example, a previous study (2016) reported low (44%) knowledge on modern contraceptive methods among Muslim women in Nepal [34]. Another study showed relatively higher (94.5%) knowledge on modern contraceptive methods in Nepal [35]. About 87% of women knew contraceptive methods in India [42]. Exposure to FP information through mass media message dissemination, community HWs and Female Community Health Volunteers (FCHVs) in the study area might have helped acquire good knowledge of modern contraceptive methods.

Despite a high proportion of good knowledge on modern contraceptive methods, Muslim women have low practices of modern contraceptive methods in Nepal. Religious beliefs, societal pressure and fear of going against religious values could be a potential driving force of lower practices of modern contraceptive methods [43]. Our study's finding is consistent with past studies conducted in Bangladesh [44], and India [45]. Injectable was the most practised modern contraceptive method, followed by oral contraceptive pills. Similar to our findings, previous research conducted in the eastern district of Nepal also reported injectables as the most used contraceptives (53.1%), followed by oral contraceptives (24%) [34]. Likewise, another study conducted in the Kapilvastu district in Nepal reported that injectable (51.3%) was the most commonly used contraceptive method, followed by oral contraceptives (25.6%) [19]. Injectables are the most preferred modern contraceptive methods among Muslim women in Nepal. Their popularity could be due to their simplicity, effectiveness for three months and accessibility even in private pharmacies at a low cost [46].

**Table 3. Factors associated with knowledge and practices of modern contraceptive methods among Muslim women (N = 400) in Nepal.**

| Variables | Frequency (%) | Knowledge modern contraceptive | | P value | Practice modern contraceptive | | P value |
|---|---|---|---|---|---|---|---|
| | | Poor (<median) (%) | Good (> = median) (%) | | No (%) | Yes (%) | |
| **Women's age** | | | | | | | |
| ≤18 years | 34 (8.5) | 12 (35.3) | 22 (64.7) | 0.805 | 15 (44.1) | 19 (55.9) | 0.580 |
| 19–29 years | 184 (46.0) | 57 (31.0) | 127 (69.0) | | 99 (53.8) | 85 (46.2) | |
| ≥30 years | 182 (45.5) | 54 (29.7) | 128 (70.3) | | 96 (52.7) | 86 (47.3) | |
| **Family type** | | | | | | | |
| Joint | 255 (63.8) | 70 (27.5) | 185 (72.5) | 0.058 | 147(57.6) | 108(42.4) | 0.006 |
| Nuclear | 145 (36.3) | 53 (36.6) | 92 (63.4) | | 63(43.4) | 82(56.6) | |
| **Parity** | | | | | | | |
| 0–2 | 314 (78.5) | 97 (30.9) | 217 (69.1) | 0.907 | 164(52.2) | 150(47.8) | 0.836 |
| ≥3 | 86 (21.5) | 26 (30.2) | 60 (69.8) | | 46(53.5) | 40(46.5) | |
| **Women's education** | | | | | | | |
| Illiterate | 108 (27.0) | 35 (32.4) | 73 (67.6) | 0.772 | 68(63.0) | 40(37.0) | 0.020 |
| Primary | 192 (48.0) | 60 (31.3) | 132 (68.8) | | 98(51.0) | 94(49.0) | |
| Secondary and above | 100 (25.0) | 28 (28.0) | 72 (72.0) | | 44(44.0) | 56(56.0) | |
| **Women's occupation** | | | | | | | |
| Housewives | 198 (49.5) | 48 (24.2) | 150 (75.8) | <0.001 | 103(52.0) | 95(48.0) | 0.444 |
| Agriculture | 158 (39.5) | 68 (43.0) | 90 (57.0) | | 80(50.6) | 78(49.4) | |
| Daily wages worker | 44 (11.0) | 7 (15.9) | 37 (84.1) | | 27(61.4) | 17(38.6) | |
| **Husband's occupation** | | | | | | | |
| Agriculture | 148 (37.0) | 53 (35.8) | 95 (64.2) | 0.078 | 79(53.4) | 69(46.6) | 0.123 |
| Business and service | 82 (20.5) | 16 (19.5) | 66 (80.5) | | 20(40.0) | 30(60.0) | |
| Daily wages workers | 50 (12.5) | 15 (30.0) | 35 (70.0) | | 40(48.8) | 42(51.2) | |
| Foreign migrant worker | 120 (30.0) | 39 (32.5) | 81 (67.5) | | 71(59.2) | 49(40.8) | |
| **Income (monthly) NRs** | | | | | | | |
| ≤20000 | 158 (39.5) | 53 (33.5) | 105 (66.5) | 0.328 | 78(49.4) | 80(50.6) | 0.311 |
| >20000 | 242 (60.5) | 70 (28.9) | 172 (71.1) | | 132(54.5) | 110(45.5) | |
| **Knowledge of modern contraceptive methods** | | | | | | | |
| Poor knowledge | | | | | 76(61.8) | 47(38.2) | 0.013 |
| Good knowledge | | | | | 134(48.4) | 143(51.6) | |
| **Ever visited a HF for FP counselling** | | | | | | | |
| No | | | | | 85 (73.3) | 31(26.7) | <0.001 |
| Yes | | | | | 125 (39.7) | 190(60.3) | |

The knowledge of modern contraceptive methods was influenced by several socioeconomic factors such as family type and occupation of women. The current study revealed that women who lived in the nuclear family and were involved in agriculture had poor knowledge of modern contraceptive methods. The women belonging to a nuclear family may have limited exposure to other family members, resulting in less opportunity to obtain information about contraceptive methods. In addition, women involved in agriculture might lack access to information on contraception. The finding of this study is consistent with the study conducted in India [47]. However, previous studies in Nepal have reported no association between the type of family and knowledge of modern contraceptive methods [34, 35].

Several determinants such as education, nuclear family, good knowledge of contraceptive methods and access to counselling services were positively associated with the practices of modern contraceptive methods. Studies from Nepal [18, 34] and other Asian countries [48]

Table 4. Determinants of good knowledge on modern contraceptive methods among Nepali Muslim women (N = 400).

| Variables | Knowledge on modern contraceptive | | | |
|---|---|---|---|---|
| | COR 95% CI | p | AOR 95% CI | p |
| **Women's age** | | | | |
| ≥30 years | 1.00 | | 1.00 | |
| ≤18 years | 0.77 (0.36,1.67) | 0.514 | 0.61 (0.25,1.48) | 0.276 |
| 19–29 years | 0.94 (0.60,1.47) | 0.786 | 0.84 (0.49,1.44) | 0.533 |
| **Family type** | | | | |
| Joint | 1.00 | | 1.00 | |
| Nuclear | 0.66 (0.42, 1.02) | 0.059 | **0.60 (0.38,0.95)** | **0.030** |
| **Parity** | | | | |
| 0–2 | 1.00 | | 1.00 | |
| ≥3 | 1.03 (0.61,1.73) | 0.907 | 1.11(0.61,2.01) | 0.728 |
| **Women's education** | | | | |
| Illiterate | 1.00 | | 1.00 | |
| Primary | 1.05 (0.64, 1.75) | 0.836 | 0.86 (0.49,1.54) | 0.620 |
| Secondary and above | 1.23 (0.68,2.23) | 0.490 | 0.77 (0.36, 1.64) | 0.495 |
| **Women's occupation** | | | | |
| Housewives | 1.00 | | 1.00 | |
| Agriculture | 0.42 (0.27,0.67) | <0.001 | **0.38 (0.22, 0.64)** | **<0.001** |
| Daily wages worker | 1.69 (0.71,4.04) | 0.237 | 1.61 (0.62,4.19) | 0.327 |
| **Husband's occupation** | | | | |
| Agriculture | 1.00 | | 1.00 | |
| Business and service | 2.30 (1.21,4.37) | 0.011 | 1.56 (0.68,3.59) | 0.295 |
| Daily wages workers | 1.30 (0.652,2.60) | 0.455 | 0.91 (0.42,1.94) | 0.802 |
| Foreign migrant worker | 1.16(0.70,1.93) | 0.570 | 0.81 (0.40,1.62) | 0.554 |
| **Income (monthly) in NRs** | | | | |
| ≤20000 | 1.00 | | 1.00 | |
| >20000 | 1.24 (0.81,1.91) | 0.328 | 1.02 (0.56, 1.83) | 0.958 |

Bold Significant at $p < 0.05$.

have reported increased practices of modern contraceptive methods with increased education [18, 48]. The findings of the current study are consistent with the previous studies conducted in Nepal [34], Bangladesh [44], and India [45]. Illiterate women may have limited access to contraceptives, leading to a lack of awareness about the benefits of contraceptive use. Furthermore, those women may not openly discuss contraceptives with their spouses due to lower autonomy in marital relationships [26, 49]. Previous evidence showed that illiterate Muslim women became unaware of their reproductive rights and were reluctant to visit health facilities for FP services [16].

Similarly, this study identified women living in the nuclear family have good practices of modern contraceptive methods despite having poor knowledge. The women in the nuclear family may be less likely to be influenced by in-laws and other family members for FP decision-making and more freedom to uptake FP services. Likewise, In the nuclear family, a supportive environment for women may have encouraged them to use family planning services despite their lack of knowledge. Future research can explore the contributing factors of low knowledge but good practices among Muslim women from the nuclear family in Nepal.

Past evidence documented that having good knowledge of contraceptive methods may increase the practice of these contraceptives [26]. Our study also showed that women's

**Table 5. Determinants of good practices of modern contraceptive methods among Nepali Muslim women (N = 400).**

| Variables | Practice of modern contraceptive | | | |
|---|---|---|---|---|
| | COR 95% CI | p | AOR 95% CI | p |
| **Women's age** | | | | |
| ≥30 years | 1.00 | | 1.00 | |
| ≤18 years | 1.41 (0.68, 2.95) | 0.357 | 1.22 (0.50,3.01) | 0.664 |
| 19–29 years | 0.96 (0.64,1.45) | 0.839 | 0.99 (0.58,1.69) | 0.968 |
| **Family type** | | | | |
| Joint | 1.00 | | 1.00 | |
| Nuclear | 1.77 (1.17, 2.67) | 0.006 | **2.24 (1.40,3.59)** | **0.001** |
| **Parity** | | | | |
| 0–2 | 1.00 | | 1.00 | |
| ≥3 | 0.95 (0.59,1.53) | 0.836 | 1.22 (0.67,2.24) | 0.511 |
| **Women's education** | | | | |
| Illiterate | 1.00 | | 1.00 | |
| Primary | 1.63 (1.01, 2.64) | 0.047 | **2.59 (1.43, 4.72)** | **0.002** |
| Secondary and above | 2.16 (1.24, 3.77) | 0.006 | **4.41(2.02,9.63)** | **<0.001** |
| **Women's occupation** | | | | |
| Housewives | 1.00 | | 1.00 | |
| Agriculture | 1.06 (0.70,1.61) | 0.795 | 1.46 (0.84,2.52) | 0.179 |
| Daily wages worker | 0.68 (0.35,1.33) | 0.263 | 0.57 (0.25,1.29) | 0.175 |
| **Husband's occupation** | | | | |
| Agriculture | 1.00 | | 1.00 | |
| Business and service | 1.72 (0.90, 3.30) | 0.104 | **2.55 (1.17,5.56)** | **0.019** |
| Daily wages workers | 1.20(0.70,2.06) | 0.504 | 1.29 (0.58,2.85) | 0.532 |
| Foreign migrant worker | 0.79 (0.49,1.29) | 0.343 | 1.00 (0.49,2.02) | 0.989 |
| **Income (monthly) in NRs** | | | | |
| ≤20000 | 1.00 | | 1.00 | |
| >20000 | 0.81 (0.54,1.21) | 0.311 | 0.67(0.36,1.21) | 0.177 |
| **Knowledge of modern contraceptive methods** | | | | |
| Poor knowledge | 1.00 | | 1.00 | |
| Good knowledge | 1.73 (1.12, 2.66) | 0.014 | **2.73 (1.66, 4.51)** | **<0.001** |
| **Ever visited a HF for FP counselling** | | | | |
| No | 1.00 | | 1.00 | |
| Yes | 3.49(2.17,5.60) | <0.001 | **4.40 (2.58,7.50)** | **<0.001** |

Bold Significant at $p < 0.05$.

knowledge of modern contraceptive methods was related to their practice. Women with good knowledge were more likely to practice modern contraceptive methods than those with poor knowledge. This might be because women with good knowledge may know better about the benefits of contraceptive use. Therefore, it would increase the women's decision making power for the practice of contraceptives [50].

Moreover, access to FP counselling was another factor affecting contraceptive practices in our study. Women who had ever visited a health facility for FP counselling were more likely to practice modern contraceptive methods than those who had not visited. Women who have ever visited a health facility for FP counselling might be aware of the benefits of contraceptive use. Therefore, they have favourable behaviour toward the practices of contraceptive methods. A similar study conducted from abroad reported consistent findings [51].

## Programmatic implications

This study has highlighted some implications for policy and programs. First, the current study revealed a satisfactory level of good knowledge and poor practices of modern contraceptive methods. These women groups require accessible quality contraceptive choices. Some targeted interventions can be adopted and implemented to improve the knowledge and practice of modern contraceptives. Such interventions include Social Behaviour Change Communication (SBCC) initiatives to raise contraceptive awareness, embedding the Muslim values and culture and mobilising health workers (HWs) from their community.

Similarly, raising awareness and developing educational materials in Urdu for Muslim women and working to extend support for smaller family norms, providing counselling and advice about the contraceptive practice from the community and religious leaders. Moreover, this study suggests that the Ministry of Health and Population (MOHP) should design targeted program strategies for Muslim women based on a deeper understanding of needs, including religious and cultural recognition.

## Strengths and limitations of the study

This study has some strengths. We have used pretested and well-designed questions and trained interviewers from the local community. The study has also explored the factors influencing the practice of modern contraceptive methods among most unreached groups. However, this study has some limitations: First, it was a survey design that did not provide us with inferences regarding causality. Second, some important covariates, such as distance to a health facility where FP service is available and cost that previous studies found important predictors of contraceptive practices, were not included in this study [52, 53]. Third, this study cannot be generalised to all populations as this study was conducted among Muslim women. Finally, though this study provided a cross-sectional analysis of knowledge and practices, a qualitative study can explore the underlying drivers of gaps in high knowledge and low practices of modern contraceptive methods among Muslim communities in Nepal.

## Conclusions

The practice of modern contraceptive methods is relatively low despite having satisfactory knowledge among Muslim women. The poor knowledge and practice of modern contraceptive methods are seen especially among socially disadvantaged groups. Therefore, improving FP practices among Nepali Muslims needs integrated and focused health interventions. Such program interventions include health education and information dissemination, SBCC interventions, and mobilisation of health workers from the Muslim community. In addition, Focusing on SBCC interventions among socially disadvantaged groups and improving access to modern contraceptive methods could improve the practices of FP services among Muslims in Nepal. Moreover, the study suggests that future studies should look into the contributing factors of low knowledge but good practices of modern contraceptive methods among Muslim women from the nuclear family in Nepal.

## Supporting information

**S1 Data.**
(SAV)

## Acknowledgments

The authors would like to acknowledge the Khajura Rural Municipality and all the participants who participated in this study.

**Disclaimer:** Views presented in this article are solely those of the authors, and do not represent views, interest, or funded work of the organisations where authors affiliated.

## Author Contributions

**Conceptualization:** Dipendra Singh Thakuri, Yamuna K. C. Singh, Resham B. Khatri.

**Data curation:** Dipendra Singh Thakuri, Yamuna K. C. Singh, Resham B. Khatri.

**Formal analysis:** Dipendra Singh Thakuri, Yamuna K. C. Singh, Resham B. Khatri.

**Investigation:** Dipendra Singh Thakuri, Yamuna K. C. Singh, Resham B. Khatri.

**Methodology:** Dipendra Singh Thakuri, Yamuna K. C. Singh, Rajendra Karkee, Resham B. Khatri.

**Software:** Dipendra Singh Thakuri, Resham B. Khatri.

**Writing – original draft:** Dipendra Singh Thakuri, Yamuna K. C. Singh, Rajendra Karkee, Resham B. Khatri.

**Writing – review & editing:** Dipendra Singh Thakuri, Yamuna K. C. Singh, Rajendra Karkee, Resham B. Khatri.

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
