## [Decision Letter · Decision Letter 0]

26 Apr 2022

PONE-D-22-08164Knowledge and practices of modern contraceptive among religious minority (Muslim) women: A cross-sectional study from Southern NepalPLOS ONE

Dear Dr. Thakuri,

Thank you for submitting your manuscript to PLOS ONE. After careful consideration, we feel that it has merit but does not fully meet PLOS ONE’s publication criteria as it currently stands. Therefore, we invite you to submit a revised version of the manuscript that addresses the points raised during the review process.

ACADEMIC EDITOR:

Results:

Was the knowledge score normally distributed? What was the cut off score for the good and poor knowledge? Please provide the results of knowledge score in mean, median and the minimum and maximum score.Table 2 Kindly change the No and yes for knowledge contraceptive methods to poor (score <xx) and good (score >= xxx).Table 3 Title: kindly specify determinants of good or poor knowledge

Pragmatic implications and conclusion

The suggestion in implication need to base on the results. This study did not show any results of male engagement, thus might not appropriate to introduce this in the pragmatic implications and conclusion.

We look forward to receiving your revised manuscript.

Kind regards,

Ai Theng Cheong

Academic Editor

PLOS ONE

Journal Requirements:

2. In the ethics statement in the Methods, you have specified that verbal consent was obtained. Please provide additional details regarding how this consent was documented and witnessed, and state whether this was approved by the IRB.

3. You indicated that you had ethical approval for your study. In your Methods section, please ensure you have also stated whether you obtained consent from parents or guardians of the minors included in the study or whether the research ethics committee or IRB specifically waived the need for their consent.

Reviewers' comments:

Reviewer's Responses to Questions

**Comments to the Author**

1. Is the manuscript technically sound, and do the data support the conclusions?

Reviewer #1: Yes

Reviewer #2: Partly

2. Has the statistical analysis been performed appropriately and rigorously? 

Reviewer #1: Yes

Reviewer #2: Yes

3. Have the authors made all data underlying the findings in their manuscript fully available?

Reviewer #1: Yes

Reviewer #2: Yes

4. Is the manuscript presented in an intelligible fashion and written in standard English?

Reviewer #1: No

Reviewer #2: No

5. Review Comments to the Author

Reviewer #1: 1. Abstract: Data were collected using face-to-face semi-structured interviews. Semi-structured is for qualitative. Remove semi.

2. In Nepal, modern contraceptive services are provided from different outlets ranging from community to tertiary level (Figure 1). If this is, what is the real gap in your study?

3. Your method part is shallow; try to address the study period, data analysis...

4. Page- 5: Method part, 1st paragraph “Muslims in Nepal”. This is unclear. What is its importance? This sensitive issue (religious-based). Better if it’s addressed in the statement of the problem as a gap by linking with your objectives.

5. Page 5: Your sample size is 417, i.e. p-0.44 + 10% non-response rate= 417. How you said 400?

6. Page 6: Conceptual framework of the study. What is the importance? i.e. its repetition you have already in the discussion part. If it has justifiable importance, take it at the end of the introduction rather than in the method part.

7. Page 6: “Study variables”. Minimize it- put the main list of the variables.

8. Page 6: Outcome variables. Similarly, minimize it- as good/poor knowledge of modern contraceptive methods is assessed from 10 items and then we considered a score of mean and above ‘Good Knowledge’ and a score of below mean ‘poor knowledge [7, 36]. Do not list everything.

9. Adopted–adaptation

10. Result part: add the descriptive finding of 10 sets of questions related to knowledge about modern contraceptives

11. It needs an English edition

Reviewer #2: This article has potential for highlighting social, religion and cultural issue in family planning among the minority ethnic group of women.

Introduction: The objectives are not clear. Investigating the status of knowledge and practice need clarification. The justification and problem gaps were not clearly laid down. Why still study the practice status when there have been studies that demonstrated the low CPR among Nepal minority group?

Methods:

Information regarding the venue not clear- work office? Was it done from the hospital base or the central office of MWRA community?

Sample size calculation needs to demonstrate a recalculation that is based on the two dependant variables (not only practice base). I would also advise to use the two proportion sample size formula.

Discussion/ conclusion: Since this study was done among a minority Muslim women; unfortunately there is very little said regarding the role of religion and culture in the family planning practice. Why would there be a low practice despite an acceptable percentage of women with good knowledge. The author earlier mentioned about the inaccessible family planning services and also being the minority ethnic group how does this influence the status of familkyplanning knowledge and practice but nothing was discussed on thesis.

This article still need English editing

6. PLOS authors have the option to publish the peer review history of their article (what does this mean?). If published, this will include your full peer review and any attached files.

Reviewer #1: **Yes: **Alemu Guta

Reviewer #2: No

---

## [Author Response · Author response to Decision Letter 0]

22 Jun 2022

June 5, 2022

Dear Editors-in-Chief

PLOS ONE 

Thank you very much for your email with the decision dated April 26, 2022. We found that reviewers' and editor feedbacks were insightful. We have addressed both reviewers’ and editors' comments point by point. We have uploaded the revised manuscript showing track changes so that you can see all revisions and modifications we have made. We believe that our revisions will satisfy you and both reviewers. We have included a clean copy and track change copy of the revised manuscript. 

Thank you for considering this manuscript for publication.

Sincerely,

Dipendra Singh Thakuri

On behalf of Yamuna KC Singh, Rajendra Karkee, Resham Bahadur Khatri 

Editor

1. Was the knowledge score normally distributed? What was the cut off score for the good and poor knowledge? Please provide the results of knowledge score in mean, median and the minimum and maximum score.

Authors response: Thank you so much for your insightful comments. The knowledge score was not normally distributed; therefore, we employed to run logistic regression to identify the factors associated with the outcome variables of this study. We dichotomised the knowledge and practices scores based on previous studies using mean cut-off point. Below the mean score, we consider as poor knowledge and mean and above as good knowledge. Additionally, we have added results of knowledge scores in mean, median and the maximum and minimum scores in the page no. 9 of revised manuscript. 

2. Table 2 Kindly change the No and yes for knowledge contraceptive methods to poor (score <xx) and good (score >= xxx).

Authors response: Thank you so much for your comments. We have changed it in the revised manuscript as suggested. 

3. Table 3 Title: kindly specify determinants of good or poor knowledge

Authors response: Thank you so much for your comments. We have revised it as suggested in the revised manuscript. “The good and poor knowledge was determined by using mean cut-off point. We considered below the mean score as poor knowledge and mean and above as good knowledge.”

4. The suggestion in implication need to base on the results. This study did not show any results of male engagement, thus might not appropriate to introduce this in the pragmatic implications and conclusion.

Authors response: Thank you so much for your important feedback. We agree with you, and we have removed it?

Reviewer #1

1. Abstract: Data were collected using face-to-face semi-structured interviews. Semi-structured is for qualitative. Remove semi.

Authors response: Thank you for your suggestion. It was a typo error. We have corrected it. 

 2. In Nepal, modern contraceptive services are provided from different outlets ranging from community to tertiary level (Figure 1). If this is, what is the real gap in your study?

Authors response: Thank you so much for your comment. Despite the availability of FP services, our study population does not go or is very reluctant to take services from local health posts and health centres.

3. Your method part is shallow; try to address the study period, data analysis...

Authors response: Thank you so much for your comments. Our study period was between June and September 2019, described under the study design and setting section on page no. 6. Similarly, we have mentioned details about data analysis under the separate data analysis section on page no.8. 

4. Page- 5: Method part, 1st paragraph “Muslims in Nepal”. This is unclear. What is its importance? This sensitive issue (religious-based). Better if it’s addressed in the statement of the problem as a gap by linking with your objectives.

Authors response: Thank you so much for your important feedback. We have reviewed and moved the information to the Introduction part. 

5. Page 5: Your sample size is 417, i.e., p-0.44 + 10% non-response rate= 417. How have you said 400?

Authors response: Thank you so much for your important comment. Non-response rate has been reported to be very low by various studies in Nepal, so we take a 5% non-response rate. Considering a 5% non-response rate, we calculated a sample size of 398, So we collected 400 as our final sample size. We have taken the reference of past studies to consider the non-response rate of 5%.

https://www.jnhrc.com.np/index.php/jnhrc/article/view/2244/939

6. Page 6: Conceptual framework of the study. What is the importance? i.e. its repetition you have already in the discussion part. If it has justifiable importance, take it at the end of the introduction rather than in the method part.

Authors response: Thank you so much for your valuable comment. We have removed it as you suggested. 

7. Page 6: “Study variables”. Minimize it- put the main list of the variables.

Authors response: Thank you so much for your suggestions. We have revised it as suggested. 

8. Page 6: Outcome variables. Similarly, minimize it- as good/poor knowledge of modern contraceptive methods is assessed from 10 items and then we considered a score of mean and above ‘Good Knowledge’ and a score of below mean ‘poor knowledge [7, 36]. Do not list everything.

Authors response: Thank you so much for your important feedback. We have revised it as suggested. Please see it in the page no. 7 of revised manuscript. 

9. Adopted–adaptation

10. Result part: add the descriptive finding of 10 sets of questions related to knowledge about modern contraceptives. 

Authors response: Thank you so much for your important comments. We have added descriptive findings of 10 sets of questions in the revised manuscript, see table 2 of the revised manuscript.

11. It needs an English edition

Authors response: Thank you so much for your comment. We have reviewed and edited the English language in the revised manuscript. 

Reviewer #2: 

This article has potential for highlighting social, religion and cultural issue in family planning among the minority ethnic group of women.

Introduction: The objectives are not clear. Investigating the status of knowledge and practice need clarification. The justification and problem gaps were not clearly laid down. Why still study the practice status when there have been studies that demonstrated the low CPR among Nepal minority group?

Authors response: Thank you so much for your comment. We have revised objectives and added information about problem gaps and justification in the introduction section.

Methods:

Information regarding the venue not clear- work office? Was it done from the hospital base or the central office of MWRA community?

Authors response: Thank you so much for your comment. It was a community based study, and data were collected at a community level. A face to face interview was carried out in the respondent’s household. We have described it in the methodology part on page no.8 of the revised manuscript. 

Sample size calculation needs to demonstrate a recalculation that is based on the two dependent variables (not only practice base). I would also advise to use the two proportion sample size formula.

Authors response: Thank you so much for your insightful comments /suggestions. We appreciate your important suggestion. We considered practice as our main outcome variable, so we only considered sample size calculation using practice prevalence. We have taken the reference of past studies where a sample was calculated based on the one dependent variable (Practice). https://bmchealthservres.biomedcentral.com/articles/10.1186/s12913-018-3643-3

http://pubs.sciepub.com/ajphr/5/1/1/

Why is the salary cut off at 20000? Why were the occupations divided as such?

Authors response: Thank you so much for your comment. We have taken reference of previously published literature for salary cut off and dividing occupations. 

https://www.ncbi.nlm.nih.gov/pmc/articles/PMC6510098/

http://www.aimspress.com/article/10.3934/publichealth.2019.3.291

Why doesn’t the "ever visit to hospital" not included inside the regression? 

Authors response: Thank you so much for your comment. We have one variable regarding “ever visited a health facility for FP counselling” that we included inside the regression. Besides that, we didn’t have variables like ever visited hospital in this study, so we couldn’t include it. 

Discussion/ conclusion: Since this study was done among a minority Muslim woman; unfortunately, there is very little said regarding the role of religion and culture in the family planning practice. 

Authors response: Thank you so much for your comments. We have added details regarding the role of religion and culture in the revised manuscript (see the introduction, pages 3 and 4). 

Why would there be a low practice despite an acceptable percentage of women with good knowledge. The author earlier mentioned about the inaccessible family planning services and also being the minority ethnic group how does this influence the status of family planning knowledge and practice, but nothing was discussed on thesis.

Authors response: Thank you so much for your comment. This mismatch between knowledge and practice shows knowledge always does not translate to practice because of other barriers, for example, cultural and economic reasons. We have discussed it in “Discussion”.

Any different characteristics between ref 31, 40, 41, 42 and this study? 

Authors response: Thank you. These are studies from different settings, i.e., Nepal, India, Bangladesh, and other countries. Ref. 31 is a study conducted among Muslim women from eastern Nepal. Furthermore, Ref. 40-42, similar studies from abroad (India, Bangladesh, and other countries), so the context of these studies differs from this study. 

This article still needs English editing

Authors response: thank you so much for your suggestion. We have reviewed and edited English.

We would like to thank editor and both the reviewers for their insightful comments and feedback. Thank you so much for inviting us to revision of this manuscript. 

Dipendra Singh Thakuri, on behalf of all co-authors

---

## [Decision Letter · Decision Letter 1]

11 Aug 2022

PONE-D-22-08164R1Knowledge and practices of modern contraceptive among religious minority (Muslim) women: A cross-sectional study from Southern NepalPLOS ONE

Dear Dr. Thakuri,

Thank you for submitting your manuscript to PLOS ONE. After careful consideration, we feel that it has merit but does not fully meet PLOS ONE’s publication criteria as it currently stands. Therefore, we invite you to submit a revised version of the manuscript that addresses the points raised during the review process.

We look forward to receiving your revised manuscript.

Kind regards,

Ai Theng Cheong

Academic Editor

PLOS ONE

Journal Requirements:

Additional Editor Comments (if provided):

Abstract

Conclusion

Suggest remove ‘Such interventions include mobilisation of health workers (HWs) from their community, and awareness of contraceptive methods embedding with values and culture of the Muslim religion.’ This would be more appropriate in discussion section.

Methodology

You have mentioned that the knowledge score is not normally distributed, if it is so, it might be more appropriate to use the median as the cut-off point. I can see that the mean and median are close i.e 6.9 vs 7, it is likely to be normally distributed, could you please check.

Table 3:

Title: From your explanation and interpretation of the results, it looks like your interest group for your knowledge outcome is ‘good knowledge’. If it so, the title should be determinants of good knowledge….., so the aOR less than 1 is considered as having poorer knowledge. Please check your interpretation and your analysis.

For value that <0.00, please change to <0.001. Kindly check through all values

Discussion

First paragraph: Please check the interpretation and the facts for those lived in nuclear family in terms of their knowledge and practice.

It is worth to add in a paragraph to discuss about those who lived in nuclear family regarding their poor knowledge but higher practice of FP.

Pragmatic implications

Could you please focus your intervention based on your results i.e. targeted the factors that could improve your practice based on your results (which is how to increase the knowledge in Muslim women, and intervention that could address this factor ‘ever visited a HF for HP counselling’).

The following facts need to be reorganised and relate it to the factors that you have found to influence your practice.

“These women groups require accessible quality of contraceptive choices. The concept of roving midwives service providers can be adopted and implemented to offer counselling and FP services at doorsteps. Second, some targeted Social Behaviour Change Communication (SBCC) can improve the

awareness of contraceptives, and practices include mass media mobilisation programs coherent with their religious values and promote them for FP services. Third, contraceptive practices can be improved through several demand and supply-side strategies. Supply-side approaches could be the recruitment of local HWs and FCHVs from the Muslim community. The local health workforce of the Muslim community can encourage them to practice contraceptives. Fourth, demand-side approaches could raise awareness and develop education materials in Urdu for Muslim women and work to extend support for smaller family norms, providing counselling and advice about the contraceptive practice from the community and religious leaders. Other approaches such as mobile camps, satellite camps, and home visits could be essential to promote the contraceptive practice. Lastly, this study suggests that the Ministry of Health and Population (MOHP) should design targeted program strategies for Muslim women based on a deeper understanding of needs, including religious and cultural recognition”

Conclusion

The following would need to reorganise or rephrase after you have tidied up the discussion on the section of pragmatic implications : ‘To improve FP practices among Muslims in Nepal needs integrated and focused health interventions. Such program interventions include health education and information dissemination, SBCC interventions, mobilisation and home visits using local midwives, advice and supplies, and male mobilisers to reach out to Muslim men. In addition, the provision of FP services in mobile and satellite health camps could improve the practices of FP services among Muslims in Nepal.’

Reviewers' comments:

Reviewer's Responses to Questions

**Comments to the Author**

1. If the authors have adequately addressed your comments raised in a previous round of review and you feel that this manuscript is now acceptable for publication, you may indicate that here to bypass the “Comments to the Author” section, enter your conflict of interest statement in the “Confidential to Editor” section, and submit your "Accept" recommendation.

Reviewer #1: All comments have been addressed

Reviewer #2: All comments have been addressed

2. Is the manuscript technically sound, and do the data support the conclusions?

Reviewer #1: Yes

Reviewer #2: Partly

3. Has the statistical analysis been performed appropriately and rigorously? 

Reviewer #1: Yes

Reviewer #2: Yes

4. Have the authors made all data underlying the findings in their manuscript fully available?

Reviewer #1: Yes

Reviewer #2: Yes

5. Is the manuscript presented in an intelligible fashion and written in standard English?

Reviewer #1: Yes

Reviewer #2: No

6. Review Comments to the Author

Reviewer #1: Thank you, the author, for your effort. The previously raised comments are addressed well. On the methods and all via out the manuscript.

Reviewer #2: Hi;

The Manuscript is much improved. However there are still a few sentences that have typo error and grammatically not correct. Some of the sentences are hanging. SO still need some work on English editing

2. There are 2 tables 3; need to redefine

3. Table 3 is cute crowded; suggest to separate the factors associated with knowledge and factors associated with practice. The narrative part can be improved by this separation.

4. The interpretation of table 4 regarding nuclear family predicting contraceptive use is inaccurate.

The discussion; sentence 4:

The practice of modern contraception is poor among women who lived in a nuclear family. But the OR for nuclear family in practising FP is 4.4

5.I feel that there is a lot to look at on how we can target women coming from nuclear family that can be elaborated.

6. Conclusion: some of the conclusions do not come from the data. such as the male mobiliser or cultural factors.

7. PLOS authors have the option to publish the peer review history of their article (what does this mean?). If published, this will include your full peer review and any attached files.

Reviewer #1: **Yes: **Alemu Guta

Reviewer #2: No

---

## [Author Response · Author response to Decision Letter 1]

14 Oct 2022

October 14, 2022

Dear Editors-in-Chief

PLOS ONE 

Thank you very much for your email with the decision dated Aug 12, 2022. We found the reviewers' and editor's feedback and comments insightful. We have carefully addressed both reviewers’ and editor’s comments point by point. To facilitate your review, we have uploaded the revised manuscript in track change and clean versions so that you can see all revisions and modifications we have made. We believe that our revisions will satisfy you and both reviewers. 

Thank you in advance for considering this manuscript for publication.

Sincerely,

Dipendra Singh Thakuri

Additional Editor Comments:

Abstract

Conclusion

Suggest remove ‘Such interventions include mobilisation of health workers (HWs) from their community, and awareness of contraceptive methods embedding with values and culture of the Muslim religion.’ This would be more appropriate in discussion section.

Authors' response: Thank you so much for your insightful suggestions. We've moved it from the conclusion section to the discussion section. (Page # 19)

Methodology

You have mentioned that the knowledge score is not normally distributed, if it is so, it might be more appropriate to use the median as the cut-off point. I can see that the mean and median are close i.e 6.9 vs 7, it is likely to be normally distributed, could you please check.

Authors response: Thank you so much for your important suggestion. We have used median as the cut-off point as advised. 

Table 3:

Title: From your explanation and interpretation of the results, it looks like your interest group for your knowledge outcome is ‘good knowledge’. If it so, the title should be determinants of good knowledge…., so the aOR less than 1 is considered as having poorer knowledge. Please check your interpretation and your analysis.

For value that <0.00, please change to <0.001. Kindly check through all values

Authors response: Thank you so much for your great comments. We have reviewed and revised it as advised. For a value that <0.00, we have changed it in the revised manuscript. 

Discussion

First paragraph: Please check the interpretation and the facts for those lived in nuclear family in terms of their knowledge and practice.

It is worth to add in a paragraph to discuss about those who lived in nuclear family regarding their poor knowledge but higher practice of FP.

Authors response: Many thanks for your suggestions. We have addressed this in the discussion part of the revised manuscript. (Page # 18-19)

Pragmatic implications

Could you please focus your intervention based on your results i.e. targeted the factors that could improve your practice based on your results (which is how to increase the knowledge in Muslim women, and intervention that could address this factor ‘ever visited a HF for HP counselling’).

The following facts need to be reorganised and relate it to the factors that you have found to influence your practice.

“These women groups require accessible quality of contraceptive choices. The concept of roving midwives service providers can be adopted and implemented to offer counselling and FP services at doorsteps. Second, some targeted Social Behaviour Change Communication (SBCC) can improve the

awareness of contraceptives, and practices include mass media mobilisation programs coherent with their religious values and promote them for FP services. Third, contraceptive practices can be improved through several demand and supply-side strategies. Supply-side approaches could be the recruitment of local HWs and FCHVs from the Muslim community. The local health workforce of the Muslim community can encourage them to practice contraceptives. Fourth, demand-side approaches could raise awareness and develop education materials in Urdu for Muslim women and work to extend support for smaller family norms, providing counselling and advice about the contraceptive practice from the community and religious leaders. Other approaches such as mobile camps, satellite camps, and home visits could be essential to promote the contraceptive practice. Lastly, this study suggests that the Ministry of Health and Population (MOHP) should design targeted program strategies for Muslim women based on a deeper understanding of needs, including religious and cultural recognition”

Authors' response: Thank you so much for your valuable comments. We have reviewed and revised this section based on our study findings. (Page# 19)

Conclusion

The following would need to reorganise or rephrase after you have tidied up the discussion on the section of pragmatic implications: ‘To improve FP practices among Muslims in Nepal needs integrated and focused health interventions. Such program interventions include health education and information dissemination, SBCC interventions, mobilisation and home visits using local midwives, advice and supplies, and male mobilisers to reach out to Muslim men. In addition, the provision of FP services in mobile and satellite health camps could improve the practices of FP services among Muslims in Nepal.’

Authors response: Thank you for your comment. We have revised it based on the revision made in the discussion on the section on pragmatic implications. (Page# 20) 

Reviewer #1: 

Thank you, the author, for your effort. The previously raised comments are addressed well. On the methods and all via out the manuscript.

Authors response: We thank the reviewer for their great comments.

Reviewer #2: 

Hi;

The Manuscript is much improved. However, there are still a few sentences that have typo error and grammatically not correct. Some of the sentences are hanging. SO still need some work on English editing

Authors response: Thank you so much for your comments. We have checked the typo error and grammar throughout the manuscript and edited English. 

2. There are 2 tables 3; need to redefine

3. Table 3 is cute crowded; suggest separating the factors associated with knowledge and factors associated with practice. The narrative part can be improved by this separation.

Authors response: Thank you so much for pointing this out. We have addressed this in the revised manuscript. Also, we have separated the knowledge and practice table as suggested, and the narrative part was revised accordingly (Page# 14-16) 

4. The interpretation of table 4 regarding nuclear family predicting contraceptive use is inaccurate.

Authors response: Thank you for your comment. We have rechecked the interpretation of table 4 regarding nuclear family and contraceptive use, and it seems fine. 

The discussion; sentence 4:

The practice of modern contraception is poor among women who lived in a nuclear family. But the OR for nuclear family in practising FP is 4.4

Authors response: Thank you so much for your important comment. It was a typo error. The women who lived in a nuclear family had good FP practice. We have corrected it in the revised manuscript. (Page #17)

5.I feel that there is a lot to look at on how we can target women coming from nuclear family that can be elaborated.

Authors' response: We have added some recommendations regarding this in the programmatic implication section of the revised manuscript.

6. Conclusion: some of the conclusions do not come from the data. such as the male mobiliser or cultural factors.

Authors response: Thank you so much for your comments. We have revised the conclusion as advised. Page #20) 

We want to thank the editor and both reviewers for their insightful comments and feedback. Thank you so much for inviting us to revise this manuscript. 

Dipendra Singh Thakuri, on behalf of all co-authors

---

## [Decision Letter · Decision Letter 2]

28 Nov 2022

Knowledge and practices of modern contraceptive among religious minority (Muslim) women: A cross-sectional study from Southern Nepal

PONE-D-22-08164R2

Dear Dr. Thakuri,

We’re pleased to inform you that your manuscript has been judged scientifically suitable for publication and will be formally accepted for publication once it meets all outstanding technical requirements.

Kind regards,

Ai Theng Cheong

Academic Editor

PLOS ONE

Additional Editor Comments (optional):

Reviewers' comments:

Reviewer's Responses to Questions

**Comments to the Author**

1. If the authors have adequately addressed your comments raised in a previous round of review and you feel that this manuscript is now acceptable for publication, you may indicate that here to bypass the “Comments to the Author” section, enter your conflict of interest statement in the “Confidential to Editor” section, and submit your "Accept" recommendation.

Reviewer #2: All comments have been addressed

2. Is the manuscript technically sound, and do the data support the conclusions?

Reviewer #2: Yes

3. Has the statistical analysis been performed appropriately and rigorously? 

Reviewer #2: Yes

4. Have the authors made all data underlying the findings in their manuscript fully available?

Reviewer #2: Yes

5. Is the manuscript presented in an intelligible fashion and written in standard English?

Reviewer #2: Yes

6. Review Comments to the Author

Reviewer #2: Dear authors, You have addressed all my comments.

The introduction puts the issue into correct perspective.

The results are arranged systematically and coherently narrated.

Nevertheless, I have a few suggestions that may make this article more meaningful.

Strength:

Please add that this study received help from the Muslim community as the interviewers and approval from the religiousleaders. This ensure higher participation from the respondents. This also could reflect in the conclusion that what these women (especially ones from the socially and economically disadvantaged group) require to practice contraception, may not be solely about having to increase knowledge but more of the support of various parties including the Muslim community and the religious leaders.

The higher practices of contraception use among women from nuclear family and those who have had visits to the healthcare reflect the empowerment these women have regarding decision making. Hence, further research is needed how they made the decisions to use contraception. A qualitative approach would be fitting to address this.

7. PLOS authors have the option to publish the peer review history of their article (what does this mean?). If published, this will include your full peer review and any attached files.

Reviewer #2: No

---

## [Editor Report · Acceptance letter]

2 Dec 2022

PONE-D-22-08164R2 

Knowledge and practices of modern contraceptives among religious minority (Muslim) women: A cross-sectional study from Southern Nepal 

Dear Dr. Thakuri:

I'm pleased to inform you that your manuscript has been deemed suitable for publication in PLOS ONE. Congratulations! Your manuscript is now with our production department. 

Kind regards, 

on behalf of

Dr. Ai Theng Cheong 

Academic Editor

PLOS ONE